

# Comparison of ventricular tachyarrhythmia recurrence between ischemic cardiomyopathy and dilated cardiomyopathy: a retrospective study

Chih-Yuan Fang[1,2], Huang-Chung Chen[1,2], Yung-Lung Chen[1,2], Tzu-Hsien Tsai[1,2], Kuo-Li Pan[2,3], Yu-Sheng Lin[2,3], Mien-Cheng Chen[1,2] and Wei-Chieh Lee[1,2]

[1] Cardiology, Kaohsiung Chang Gung Memorial Hospital, Kaohsiung, Taiwan
[2] Chang Gung University College of Medicine, Taiwan
[3] Cardiology, Chang Gung Memorial Hospital, Chiayi, Taiwan

## ABSTRACT

**Background:** The use of an implantable cardioverter-defibrillator (ICD) has been established as an effective secondary prevention strategy for ventricular tachycardia (VT)/ventricular fibrillation (VF). However, few reports discuss the difference in clinical predictors for recurrent VT/VF between patients with ischemic cardiomyopathy (ICM) and patients with dilated cardiomyopathy (DCM).

**Methods:** From May 2004 to December 2015, 132 consecutive patients who had ICM ($n = 94$) or DCM ($n = 38$) and had received ICD implantation for secondary prevention were enrolled in this study. All anti-tachycardia events during follow-up were validated. The clinical characteristics and echocardiographic parameters were obtained for comparison. The incidence of recurrence of VT/VF, cardiovascular mortality, all-cause mortality, the change of left ventricular ejection fraction (LVEF) and LV volume were analyzed.

**Results:** At a mean follow-up of $3.62 \pm 2.93$ years, 34 patients (36.2%) in the ICM group and 22 patients (57.9%) in the DCM group had a recurrence of VT/VF episodes ($p = 0.032$). The DCM group had a lower LVEF ($p = 0.019$), a larger LV end-diastolic volume (LVEDV) ($p = 0.001$), a higher prevalence of LVEDV >158 mL ($p = 0.010$), and a larger LV end-systolic volume ($p = 0.010$) than the ICM group. LVEDV >158 mL and no use of angiotensin-converting-enzyme inhibitor/ angiotensin receptor blocker were independent predictors of recurrences of VT/VF in ICM patients but not in DCM patients. There were no difference in cardiovascular mortality and all-cause mortality between the ICM and DCM patients.

**Conclusion:** The DCM patients had a higher recurrence rate of VT/VF than did the ICM patients during long-term follow-up. An enlarged LV is an independent predictor of the recurrence of VT/VF in ICM patients receiving ICD for secondary prevention.

Corresponding authors
Mien-Cheng Chen,
chenmien@ms76.hinet.net
Wei-Chieh Lee,
leeweichieh@yahoo.com.tw

**Subject** Cardiology
**Keywords** Implantable cardioverter-defibrillator, Ventricular tachyarrhythmia, Ischemic cardiomyopathy, Dilated cardiomyopathy

## INTRODUCTION

Cardiac arrhythmias impose a public health and an economic burden on the global medical community. Ventricular tachyarrhythmias are significantly associated with increased risks of cardiovascular complications and sudden cardiac death (SCD), consequently leading to a decreased quality of life and increased disability, high mortality, and greater healthcare expenses. In Asia, SCD occurs in approximately 40 cases per 100,000 individuals annually, and most cases of SCD are caused by myocardial infarction and ventricular tachycardia (VT)/ventricular fibrillation (VF) (*Murakoshi & Aonuma, 2013*). Implantation of an implantable cardioverter-defibrillator (ICD) has been established as an effective secondary prevention strategy for SCD, and the number of ICD implantations has increased gradually because more and more patients with post-myocardial infarction and heart failure (HF) survive with contemporary optimal medical therapies, including ß-blockers, renin-angiotensin-aldosterone antagonists, and statins, as well as modifications of risk factors (*Borne et al., 2017*). According to the current European Society of Cardiology guidelines (*Ponikowski et al., 2016*) one primary prevention ICD, the patients with non-ischemic disease fulfill indications of ejection fraction (EF) ≤35% on optimal medical therapy and with >1-year life expectancy, and the patients with ischemic disease fulfill indications when >6 weeks after MI, with EF ≤35% on optimal medical therapy and with >1-year life expectancy. Secondary prevention ICD refers to the prevention of SCD in patients who have survived a prior sudden cardiac arrest or a sustained VT (*Kusumoto et al., 2014*). Although randomized control studies demonstrated a survival benefit of ICD implantation among patients surviving SCD, the overall morbidity and mortality in this population remain high. In recent large registry reports, the survival rate for ICD-treated patients was near 90% at 1-year follow-up, and most of the deaths were related to cardiac causes (*Katz et al., 2017*). The most common causes of deaths in patients with HF include recurrent VT/VF and HF progression (*Narang et al., 1996*).

A left ventricular ejection fraction (LVEF) <40%, permanent atrial fibrillation, and QRS duration >150 m/s have been reported as independent predictors for recurrence of VT/VF in patients with dilated cardiomyopathy (DCM) (*Klein et al., 2006*). A previous study also reported left ventricular (LV) remodeling and a QRS width >125 m/s to be independent predictors of VT/VF recurrence in ICD recipients for secondary prevention under optimal medical therapy (*Lee et al., 2016*). However, few reports have focused on the differences in clinical predictors for recurrent VT/VF after receiving ICD for secondary prevention between patients with ischemic cardiomyopathy (ICM) and patients with DCM.

Accordingly, this study aimed to investigate the difference in predictors for recurrent VT/VF after ICD implantation between patients with ICM and patients with non-ischemic DCM.

## METHODS

### Database

The protocol was set according to our previous work examining predictors for recurrent VT/VF in secondary prevention ICD recipients (*Lee et al., 2016*). The type of data collected

was also similar to that in the above study. Specifically, the study extended to the follow-up period and enrolled more patients in our hospital. In addition, the study focused on the comparison of recurrent VT/VF between the ICM and the DCM groups. Recurrent VT/VF was defined as a sustained VT (duration longer than 30 s) and VF. Baseline characteristics such as general demographics, heart diseases, comorbidities, the LV function, the functional class of HF, QRS length, primary presenting rhythm, systolic blood pressure (SBP), renal function, medication, VT/VF detection zone, and VT ablation were compared between the two groups. According to the chart review, data on cardiovascular death and all-cause death were collected and compared between the groups. All patients who received ICD implantation had regular out-patient department follow-ups and underwent ICD record follow-up every three months in our hospital.

## Patient population (inclusion and exclusion criteria)

From May 2004 to December 2015, 132 consecutive patients, who had survived sudden death related to VT/VF events, were diagnosed with ICM or DCM, and received ICD implantation for secondary prevention were enrolled in this study in Kaohsiung Chang Gung Memorial Hospital. We excluded the patients who received implantable cardiac resynchronization therapy defibrillator implantation and those with other etiologies, such as Brugada syndrome, idiopathic VF or arrhythmogenic right ventricular cardiomyopathy receiving ICD implantation. The ICM group comprised 94 patients, and the non-ischemic DCM group comprised 38 patients. In the DCM group, all patients underwent coronary angiography to exclude obstructive coronary lesions. Only seven patients had focal 50% stenotic lesions in the coronary artery that did not involve the left anterior descending artery. All patients received the echocardiography evaluation at the registry inclusion prior to ICD implantation. All patients were also administered guideline-base treatments for ventricular tachyarrhythmia and HF if the patient could tolerate without decrease of renal function. All information from ICD integgoration during follow-up or anti-tachycardia events were reviewed and validated with the occurrences of VT/VF and anti-tachycardia therapy (anti-tachycardia pacing or shock) by two different electrophysiologists independently.

## Echocardiography

Echocardiographic parameters, including LV diastolic dysfunction, LV end-diastolic volume (LVEDV), and LV end-systolic volume (LVESV), were measured using a Philips IE33 or GE's Vivid 9. LVEDV and LVESV were quantified by M-mode and corrected by the two-dimensional guided biplane Simpson's method of disc measurements by echocardiography (*Crawford et al., 1980*; *Lang et al., 2005*).

## Study endpoints

The primary study endpoints included the recurrence of sustained VT/VF (longer than 30 s) which needed anti-tachycardia pacing therapy or ICD shock therapy. The secondary endpoints included cardiovascular death (death related to HF and arrhythmic death)

and all-cause mortality from any cause (including cardiovascular death, sepsis, hepatic failure, and brain hemorrhage).

## Ethics statement

The study protocol conforms to the ethical guidelines of the 1975 Declaration of Helsinki and was approved by the Institutional Review Committee for Human Research (201701405B0) of our institution. The raw data were from the ICD registry of Kaohsiung Chang Gung Memorial Hospital. The accession number for the KCGMH ICD registry was 104-8143B, and it was deposited at the Kaohsiung Chang Gung Memorial Hospital.

## Statistical analysis

Data are presented as mean ± standard deviation or percentages; median and interquartile range were used for non-normally distributed parameters. The clinical characteristics of the study groups were compared by the $t$-test or Mann–Whitney $U$ test for continuous variables or chi-square test or Kruskal–Wallis test for categorical variables. The significant predictors for the recurrence of VT/VF after ICD implantation were identified by the univariate and multivariate Cox regression analyses. Each independent variable was based on previous studies and conventional risk factors, and predictors for the recurrence of VT/VF were expressed as hazard ratios with 95% confidence intervals. Receiver operating characteristic (ROC) curves were used to determine the optimal values in terms of sensitivity and specificity. The Kaplan–Meier method and log-rank test were used to compare the event-free survival of the recurrence of VT/VF, cardiovascular mortality, and all-cause mortality during follow-up. Statistical analysis was carried out using statistical software (SPSS for Windows, Version 22; SPSS, Inc., Chicago, IL, USA). A two-sided $p$-value of 0.05 was considered statistically significant.

# RESULTS

## Receiver operating characteristic curves

Receiver operating characteristic curves for LVEDV were constructed, and they revealed that the cut-off point for the LVEDV was 158 mL. This resulted in the best sensitivities and specificities of recurrent VT/VF in the ICM group; the areas under these curves was 0.694 ($p = 0.002$). In the DCM group, ROC curves for LVEDV did not have significant values for recurrent VT/VF.

## Baseline characteristics of study patients

A total of 94 patients with a mean age of 66.7 ± 10 years were in the ICM group, and the majority was male (77.7%). A total of 38 patients with a mean age of 59.7 ± 12 years were in the non-ischemic DCM group and the majority was male (78.9%). The ICM group contained a statistically significant number of older patients and had a significantly higher prevalence of coronary artery disease, hypertension, diabetes mellitus, and hyperlipidemia than did the ICM group (Table 1). The HF functional class was similar between the two groups. The majority of primary presenting rhythm was VT in the ICM group, and the majority of the primary presented rhythm was VT plus VF in the DCM

| Table 1 Baseline characteristics of study patients. | | | |
|---|---|---|---|
| | **ICM (n = 94)** | **DCM (n = 38)** | **p-Value** |
| **General demographics** | | | |
| Age (year) | 67.7 ± 10 | 59.6 ± 11 | 0.001 |
| Male gender | 73 (77.7) | 30 (78.9) | 0.871 |
| **Comorbidity** | | | |
| CAD | 94 (100) | 7 (18.4) | <0.001 |
| Valvular heart disease post operation | 6 (6.4) | 3 (7.9) | 0.716 |
| Hypertension | 69 (73.4) | 20 (52.6) | 0.025 |
| Diabetes mellitus | 41 (43.6) | 7 (18.4) | 0.009 |
| Prior stroke | 14 (14.9) | 4 (10.5) | 0.588 |
| Hyperlipidemia | 39 (41.5) | 5 (13.2) | 0.002 |
| ESRD | 14 (14.9) | 1 (2.6) | 0.066 |
| CKD stage ≥3 | 25 (26.6) | 4 (10.5) | 0.043 |
| **Atrial fibrillation (%)** | | | 0.135 |
| No | 73 (77.7) | 23 (60.5) | |
| Paroxysmal | 14 (14.9) | 10 (26.3) | |
| Persistent | 7 (7.4) | 5 (13.2) | |
| **Heart failure** | | | 0.189 |
| NYHA functional class I–II | 63 (67.0) | 24 (63.2) | |
| NYHA functional class III–IV | 31 (33.0) | 14 (36.8) | |
| **QRS length (m/s)** | 110.0 (92.0–126.0) | 112.0 (96.0–152.0) | 0.330 |
| **Primary presenting rhythm** | | | <0.001 |
| VT | 82 (87.2) | 9 (23.7) | |
| VF | 7 (7.4) | 1 (2.6) | |
| VT plus VF | 5 (5.3) | 28 (73.7) | |
| **Systolic blood pressure (mmHg)** | 111.84 ± 14.10 | 117.45 ± 11.80 | 0.021 |
| **Creatinine (mg/dL)** | 1.31 (0.97–2.10) | 1.13 (0.90–1.63) | 0.128 |
| **Medications** | | | |
| ACEI/ARB | 66 (70.2) | 36 (94.7) | 0.002 |
| ß-blocker | 59 (62.8) | 27 (71.1) | 0.423 |
| Diuretic | 28 (29.8) | 16 (42.1) | 0.221 |
| Statin | 50 (53.2) | 6 (15.8) | <0.001 |
| Spironolactone | 16 (17.0) | 11 (28.9) | 0.154 |
| Anti-platelet agent | 85 (90.4) | 9 (23.7) | <0.001 |
| Warfarin | 11 (11.7) | 8 (21.1) | 0.179 |
| NOAC | 2 (2.1) | 2 (5.3) | 0.578 |
| Amiodarone | 70 (74.5) | 26 (68.4) | 0.521 |
| **ICD chamber** | | | 0.850 |
| Single | 50 (53.2) | 21 (55.3) | |
| Dual | 44 (46.8) | 17 (44.7) | |
| **Lowest VT-detection zone (bpm)** | 160.0 (150.0–167.0) | 164.5 (160.0–180.0) | 0.028 |
| **Lowest VF-detection zone (bpm)** | 200.0 (200.0–214.0) | 200.0 (200.0–222.0) | 0.786 |

(Continued)

| | ICM (*n* = 94) | DCM (*n* = 38) | *p*-Value |
|---|---|---|---|
| **Post VT ablation (%)** | 8 (8.5) | 6 (15.8) | 0.226 |
| Success | 4 (50) | 2 (33.3) | |
| Failure | 4 (50) | 4 (66.7) | |
| **LV systolic function** | | | |
| LVEF (%) | 44.40 ± 15.32 | 37.46 ± 14.57 | 0.019 |
| LVEDV (mL) | 167.0 (124.0–201.0) | 209.0 (167.0–264.0) | 0.001 |
| LVEDV >158 mL | 50 (53.2) | 29 (78.4) | 0.010 |
| LVESV (mL) | 85.5 (59.0–130.0) | 121.0 (74.8–168.5) | 0.010 |
| **Recurrent VT/VF (%)** | 34 (36.2) | 22 (57.9) | 0.032 |
| **VT/VF occurrence within 1 year** | 21 (26.6) | 12 (35.3) | 0.373 |
| **1-year CV mortality** | 5 (6.3) | 3 (9.4) | 0.687 |
| **1-year all-cause mortality** | 10 (11.8) | 5 (14.7) | 0.761 |
| **Follow-up time (years)** | 3.03 (1.33–5.70) | 3.01 (1.01–6.06) | 0.966 |

Notes:
Data are expressed as mean ± SD or median (IQR) if non-normal distributed parameters or as number (percentage).
ICM, ischemic cardiomyopathy; DCM, dilated cardiomyopathy; CAD, coronary artery disease; ESRD, end stage renal disease; CKD, chronic kidney disease; NYHA, New York Heart Association; ACEI, angiotensin-converting-enzyme inhibitor; ARB, angiotensin receptor blocker; NOAC, non-vitamin K oral anticoagulants; ICD, Implantable cardioverter-defibrillator; VT, ventricular tachycardia; VF, ventricular fibrillation; LVEF, left ventricular ejection fraction; LVEDV, left ventricular end diastolic volume; LVESV, left ventricular end systolic volume; CV, cardiovascular.

group ($p < 0.001$). SBP was well controlled in both groups and was higher in the DCM group. The prevalence of the use of angiotensin-converting-enzyme inhibitor (ACEI)/ angiotensin receptor blocker (ARB) was significantly lower and the prevalence of the use of statin, and anti-platelet agents was significantly higher in the ICM group compared to the DCM group. Only eight (8.5%) patients in the ICM group and six (15.8%) patients in the DCM group received VT ablation ($p = 0.226$). The DCM group had a lower LVEF (37.46 ± 14.57% vs. 44.40 ± 15.32%; $p = 0.019$), a larger LVEDV (209.0 (167.0–264.0) vs. 167.0 (124.0–201.0) mL; $p = 0.001$), higher prevalence of LVEDV >158 mL (78.4% vs. 53.2%; $p = 0.010$), and a larger LVESV (121.0 (74.8–168.5) vs. 85.5 (59.0–130.0) mL; $p = 0.010$) than the ICM group.

## Differences in the incidence of recurrent VT/VF, cardiovascular mortality, and all-cause mortality during follow-up between ICM and DCM patients

The median number of follow-up years was similar between the two groups (3.03 (1.33–5.70) years vs. 3.01 (1.01–6.06) years; $p = 0.966$). The DCM group had a significantly higher incidence of recurrent VT/VF (57.9% vs. 36.2%; $p = 0.032$) than the ICM group. The difference in event-free survival from VT/VF occurrence between the ICM and DCM patients began to separate at the 3-year follow-up and became significant at the 5-year follow-up (57.7% vs. 84.6%; $p = 0.022$) (Fig. 1).

There were no significant differences in cardiovascular mortality and all-cause mortality between the two groups at the 1-year and 3-year follow-ups (Table 1). Five patients

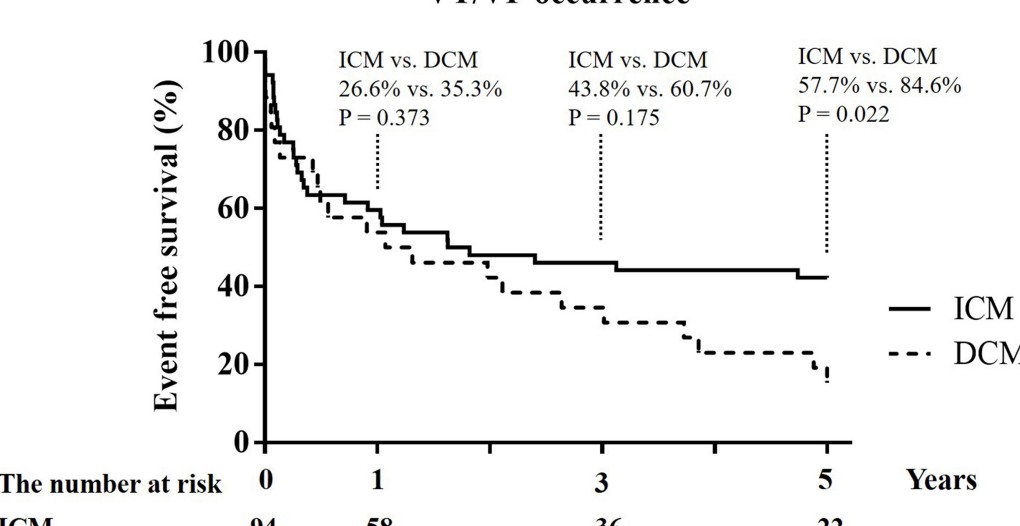

**Figure 1 Kaplan–Meier curves of 1-year, 3-year, and 5-year event free survival from recurrent ventricular tachycardia (VT)/ventricular fibrillation (VF) between ICM and DCM.** Study sites: ICM, ischemic cardiomyopathy; DCM, dilated cardiomyopathy.

in the ICM group had cardiovascular deaths due to refractory VT/VF, and three patients in the DCM group experienced cardiovascular death (two patients because of severe HF with cardiogenic shock and one patient because of refractory VT/VF) at the 1-year follow-up (Table 1).

## Clinical predictors of recurrence of VT/VF in patients with ICM

On univariate Cox regression analyses, larger LVEDV, LVEDV >158 mL, non-use of ACEI/ARB, and use of spironolactone were found to be statistically significant predictors of recurrence of VT/VF in patients with ICM (Table 2). By multivariate Cox regression analyses of LVEDV >158 mL, no use of ACEI/ARB, and the use of spironolactone, LVEDV >158 mL and no use of ACEI/ARB were independent predictors of recurrence of VT/VF in patients with ICM (Table 2).

## Clinical predictors of recurrence of VT/VF in patients with DCM

By univariate Cox regression analyses, age, gender, atrial fibrillation, LV function, LV volume, clinical HF functional class, QRS length, and medications (ACEI/ARB, ß-blocker, amiodarone, spironolactone) were not predictors of recurrence of VT/VF in patients with DCM (Table 3).

## Comparison of the recurrent rate of VT/VF between DCM patients and ICM patients with LV volume and function like DCM patients

A total of 50 ICM patients had LVEDV >158 mL and the mean LV function and volume like the 38 DCM patients. These 50 ICM patients were older and had less use

**Table 2 Univariate and multivariate Cox regression analyses in predicting recurrence of VT/VF in patients with ICM.**

| Variables | Univariate analysis | | | Multivariate analysis | | |
|---|---|---|---|---|---|---|
| | Hazard ratio | 95% CI | P value | Hazard ratio | 95% CI | p-Value |
| Female | 0.733 | 0.340–1.581 | 0.429 | | | |
| Age | 0.979 | 0.948–1.011 | 0.189 | | | |
| Atrial fibrillation (paroxysmal and persistent) | 0.678 | 0.261–1.759 | 0.424 | | | |
| LVEF (%) | 0.990 | 0.968–1.014 | 0.415 | | | |
| LVEF ≤30% | 1.674 | 0.723–3.877 | 0.229 | | | |
| LVEDV (mL) | 1.005 | 1.000–1.009 | 0.048 | | | |
| LVEDV >158 mL | 4.146 | 1.708–10.065 | 0.002 | 4.011 | 1.648–9.759 | 0.002 |
| LVESV (mL) | 1.003 | 0.998–1.009 | 0.200 | | | |
| Heart failure NYHA functional class ≥3 | 1.001 | 0.485–2.066 | 0.999 | | | |
| QRS width (m/s) | 1.002 | 0.990–1.014 | 0.704 | | | |
| ACEI/ARB | 0.448 | 0.224–0.897 | 0.023 | 0.486 | 0.239–0.959 | 0.038 |
| ß-blocker | 0.689 | 0.345–1.377 | 0.292 | | | |
| Amiodarone | 0.768 | 0.372–1.585 | 0.475 | | | |
| Spironolactone | 2.261 | 1.002–5.103 | 0.049 | | | |

**Note:**
ICM, ischemic cardiomyopathy; VT, ventricular tachycardia, VF, ventricular fibrillation, CI, confidence interval, LVEF, left ventricular ejection fraction, LVEDV, left ventricular end diastolic volume, LVESV, left ventricular end systolic volume, NYHA, New York Heart Association, ACEI, angiotensin-converting-enzyme inhibitor; ARB, angiotensin receptor blocker.

of ACEI/ARB than the 38 DCM patients (Table 4) and there was no difference in the clinical functional class of HF, QRS length, the prevalence of use of ß-blocker, diuretics, spironolactone, and amiodarone between the 50 ICM patients and 38 DCM patients. The 1-year recurrence rate of VT/VF (34.0% vs. 35.3%; $p = 1.000$), 1-year cardiovascular mortality, and 1-year all-cause mortality did not differ between these 50 ICM patients and the 38 DCM patients (Table 4; Fig. 2).

## The incidence of recurrent ventricular tachyarrhythmia in the patients with improving LVEDV or LVEF at 1-year follow-up period

After excluding the expired patients and the patients who did not receive the follow-up echocardiography between the half-year and 1-year follow-up period, a total the 101 patients received follow-up echocardiography after at least half a year of medical treatment. LVEDV regressed ≥10% with less prevalence of recurrent ventricular tachyarrhythmia but did not reach significance (Table 5). LVEDV still greater than 158 mL had a significantly higher prevalence of recurrent ventricular tachyarrhythmia (LVEDV >158 vs. ≤158 mL; 54.2% vs. 31.0%; $p = 0.026$) (Table 5). The patients with improving LVEF greater than 5% did not have a lesser prevalence of recurrent ventricular tachyarrhythmia (Table 5).

**Table 3 Univariate Cox regression analyses in predicting recurrence of VT/VF in patients with DCM.**

| Variables | Univariate analysis | | |
|---|---|---|---|
| | Hazard ratio | 95% CI | *p*-Value |
| Female | 1.078 | 0.363–3.205 | 0.892 |
| Age | 1.007 | 0.971–1.043 | 0.717 |
| Atrial fibrillation (paroxysmal and persistent) | 1.563 | 0.665–3.674 | 0.306 |
| LVEF (%) | 1.001 | 0.968–1.036 | 0.947 |
| LVEF $\leq$30% | 1.326 | 0.551–3.191 | 0.529 |
| LVEDV (mL) | 1.003 | 0.998–1.009 | 0.253 |
| LVEDV >158 mL | 1.641 | 0.549–4.907 | 0.376 |
| LVESV (mL) | 1.005 | 0.999–1.010 | 0.123 |
| Heart failure NYHA functional class $\geq$3 | 0.870 | 0.337–2.243 | 0.773 |
| QRS width (m/s) | 1.009 | 0.996–1.022 | 0.166 |
| ACEI/ARB | 0.369 | 0.083–1.634 | 0.189 |
| ß-blocker | 0.481 | 0.176–1.313 | 0.153 |
| Amiodarone | 1.630 | 0.653–4.068 | 0.296 |
| Spironolactone | 0.509 | 0.172–1.506 | 0.223 |

**Note:**

DCM, dilated cardiomyopathy; VT, ventricular tachycardia, VF, ventricular fibrillation, CI, confidence interval, LVEF, left ventricular ejection fraction, LVEDV, left ventricular end diastolic volume, LVESV, left ventricular end systolic volume, NYHA, New York Heart Association, ACEI, angiotensin-converting-enzyme inhibitor; ARB, angiotensin receptor blocker.

## DISCUSSION

This study showed that the DCM patients had a significantly higher incidence of recurrent VT/VF than did the ICM patients. In both groups, SBP was well controlled, and a guideline-based treatment was applied if possible. Besides, LVEDV >158 mL and non-use of ACEI/ARB were independent predictors of recurrence of VT/VF in the patients with ICM but not in the DCM patients. However, like DCM patients, the subgroup of ICM patients with LV function and volume had a recurrence rate of VT/VF that was similar to that of DCM patients. There were no differences in the cardiovascular mortality and all-cause mortality between the ICM patients and DCM patients. After adequate medical treatment for at least half a year, a higher prevalence of recurrent ventricular tachyarrhythmia was noted in the patients with LVEDV >158 mL.

One study reported that around 50% of recurrent VT/VF events developed in DCM patients with ICD implantation for primary prevention and secondary prevention, and the time between implantation and first appropriate therapy was similar between primary and secondary prevention patients (*Karaoguz et al., 2006*; *Meyer et al., 2009*). Although there are controversial reports regarding the use of ICD implantation to reduce sudden death in patients with DCM (*Anderson, 2005*), few studies have discussed the clinical predictors of recurrent VT/VF in DCM patients. In this study, LVEDV >158 mL was an independent

**Table 4 Comparison of the recurrent of VT/VF between DCM patients and ICM patients with LV volume and function like DCM patients.**

| | ICM (n = 50) | DCM (n = 38) | p-Value |
|---|---|---|---|
| **General demographics** | | | |
| Age (year) | 65.1 ± 11 | 59.6 ± 11 | 0.031 |
| Male gender | 41 (82.0) | 30 (78.9) | 0.789 |
| **Heart failure** | | | 0.827 |
| NYHA functional class I–II | 30 (60.0) | 24 (63.2) | |
| NYHA functional class III–IV | 20 (40.0) | 14 (36.8) | |
| **QRS length (m/s)** | 110.0 (102.0–120.0) | 112.0 (96.0–152.0) | 0.514 |
| **Medications** | | | |
| ACEI/ARB | 34 (68.0) | 36 (94.7) | 0.003 |
| ß-blocker | 30 (60.0) | 27 (71.1) | 0.369 |
| Diuretic | 23 (46.0) | 16 (42.1) | 0.829 |
| Spironolactone | 12 (24.0) | 11 (28.9) | 0.631 |
| Amiodarone | 37 (74.0) | 26 (68.4) | 0.636 |
| **LV systolic function** | | | |
| LVEF (%) | 38.82 ± 13.33 | 37.46 ± 14.57 | 0.652 |
| LVEDV (mL) | 201.0 (176.5–255.0) | 209.0 (167.0–264.0) | 0.859 |
| LVESV (mL) | 127.0 (104.0–176.5) | 121.0 (74.8–168.5) | 0.756 |
| **Recurrent VT/VF (%)** | 24 (48.0) | 22 (57.9) | 0.395 |
| **VT/VF occurrence within 1 year** | 17 (34.0) | 12 (35.3) | 1.000 |
| **1-year CV mortality** | 3 (6.0) | 3 (9.4) | 0.688 |
| **1-year all-cause mortality** | 4 (8.0) | 5 (14.7) | 0.484 |

Notes:
Data are expressed as mean ± SD or median (IQR) if non-normal distributed parameters or as number (percentage).
VT, ventricular tachycardia; VF, ventricular fibrillation; ICM, ischemic cardiomyopathy; DCM, dilated cardiomyopathy; LVEDV, left ventricular end diastolic volume; NYHA, New York Heart Association; ACEI, angiotensin-converting-enzyme inhibitor; ARB, angiotensin receptor blocker; LV, left ventricle; LVEF, left ventricular ejection fraction; LVESV, left ventricular end systolic volume; CV, cardiovascular.

predictor for recurrent VT/VF in patients with ICM, but not in patients with DCM. An item of note is that, like DCM patients, ICM patients with LVEDV >158 mL and LV volume had a recurrence rate of VT/VF similar to that of DCM patients. LV dilatation has been reported as a major negative prognostic marker in patients with HF (*Frigerio & Roubina, 2005*) and indicates a diseased heart with a high possibility of recurrent ventricular tachyarrhythmia. In this study, a higher prevalence of recurrent ventricular tachyarrhythmia was noted in dilated LVEDV before and after ICD implantation and after adequate treatment. Although ICD therapy could prevent SCD, ICD could not halt or reverse the progression of HF. Therefore, optimal medical therapy, especially the use of ACEI/ARB to halt or reverse LV remodeling and HF progression is crucial to reduce the recurrence of VT/VF in ICM patients.

A previous study reported that the survival of patients undergoing appropriate ICD therapy was 55% at the 4-year follow-up, and the survival of patients with all-cause

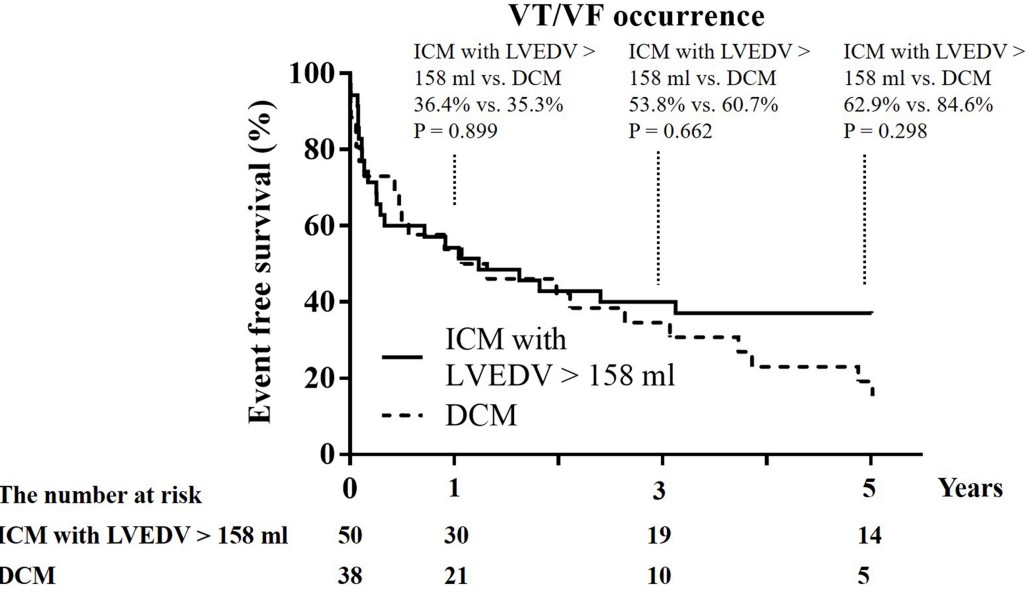

**Figure 2 Kaplan–Meier curves of 1-year, 3-year, and 5-year event free survival from recurrent ventricular tachycardia (VT)/ventricular fibrillation (VF) between ICM with LVEDV >158 mL and DCM.** Study sites: ICM, ischemic cardiomyopathy; DCM, dilated cardiomyopathy; LVEDV, left ventricular end diastolic volume.

**Table 5 The incidence of recurrent ventricular tachyarrhythmia in the patients with improving left ventricular diastolic volume or left ventricular ejection fraction at 1-year follow-up period.**

|  |  | The incidence of ventricular tachyarrhythmia | p-Value |
|---|---|---|---|
| LVEDV | Smaller ≥10% | 16/41 (39.0%) | 0.417 |
|  | Smaller <10% | 29/60 (48.3%) |  |
| LVEDV | ≥158 mL | 32/59 (54.2%) | 0.026 |
|  | <158 mL | 13/42 (31.0%) |  |
| LVEF | Improving ≥5% | 20/43 (46.5%) | 0.840 |
|  | Improving <5% | 25/58 (43.1%) |  |

**Note:**
LVEDV, left ventricular end diastolic volume; LVEF, left ventricular ejection fraction.

mortality was 79% in DCM patients receiving ICD for secondary prevention (*Karaoguz et al., 2006*). In this study, we found that there were no differences in cardiovascular mortality and all-cause mortality between the ICM and DCM patients receiving ICD implantation for secondary prevention. These may be contributed by the differences in underlying disease with higher prevalence of diabetes mellitus, chronic kidney disease stage ≥3 and hypertension, and less guideline-based treatments in ICM patients.

Left ventricular remodeling and dilation developed due to myocardial infarction and contributed to LV contractile dysfunction in the survivors of acute myocardial infarction. LV dilation after myocardial infarction is an important determinant of future major adverse cardiovascular events including ventricular arrhythmias, HF, and death (*St. John Sutton et al., 1997*). The European Society of Cardiology 2016 guidelines recommended

ACEI as an anti-arrhythmia agent to decrease ventricular tachyarrhythmia in patients with HF and reduced LVEF (*Ponikowski et al., 2016*). ACEI may not be a traditional antiarrhythmic agent, but ACEI has been shown to reverse post-infarction LV remodeling, alter LV architecture and function, and decrease the development of essential substrates for high-grade ventricular arrhythmias (*St. John Sutton et al., 2003*). ACEI cannot be qualified as an antiarrhythmic agent by any means, but the use of a neurohormonal blockade can reduce the incidence of VT through reverse remodeling. In our study, non use of ACEI/ARB was found to be an independent predictor of recurrent VT/VF. In clinical practice, decreased renal function or increased disease severity and hypotension often limit the addition of ACEI/ARB. However, the results of our study provided important evidence on the use of ACEI/ARB to decrease ventricular tachyarrhythmia by decreasing LV remodeling.

### Study limitations

This retrospective analysis had several limitations. First, this study population was relatively small. However, we provided important information in terms of the difference in clinical predictors for recurrent VT/VF, after receiving ICD for secondary prevention, between patients with ICM and patients with non-ischemic DCM. Second, electrocardiographic variability and biomarkers were not included in the analysis, and this study included only conventional clinical parameters.

## CONCLUSION

The non-ischemic DCM patients had a higher recurrence rate of VT/VF than the ICM patients during long-term follow-up. An enlarged LV is an independent predictor of recurrence of VT/VF in ICM patients who received ICD for secondary prevention. The non-ischemic DCM patients and the ICM patients had similar long-term cardiovascular mortality and all-cause mortality.

### Funding

The authors received no funding for this work.

### Competing Interests

The authors declare that they have no competing interests.

### Author Contributions

- Chih-Yuan Fang conceived and designed the experiments, authored or reviewed drafts of the paper, approved the final draft.
- Huang-Chung Chen performed the experiments, authored or reviewed drafts of the paper, approved the final draft.
- Yung-Lung Chen performed the experiments, authored or reviewed drafts of the paper, approved the final draft.

![PeerJ]

- Tzu-Hsien Tsai performed the experiments, authored or reviewed drafts of the paper, approved the final draft.
- Kuo-Li Pan performed the experiments, authored or reviewed drafts of the paper, approved the final draft.
- Yu-Sheng Lin performed the experiments, authored or reviewed drafts of the paper, approved the final draft.
- Mien-Cheng Chen performed the experiments, analyzed the data, authored or reviewed drafts of the paper, approved the final draft.
- Wei-Chieh Lee conceived and designed the experiments, performed the experiments, analyzed the data, contributed reagents/materials/analysis tools, prepared figures and/or tables, authored or reviewed drafts of the paper, approved the final draft.

## Human Ethics

The following information was supplied relating to ethical approvals (i.e., approving body and any reference numbers):

The study protocol conforms to the ethical guidelines of the 1975 Declaration of Helsinki and was approved by the Kaohsiung Chang Gung Memorial Hospital's Institutional Review Committee for Human Research (201701405B0).

## Data Availability

The raw data are provided in a Supplemental File.

## Supplemental Information

Supplemental information for this article can be found online at http://dx.doi.org/10.7717/peerj.5312#supplemental-information.

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
