# Peer review of "Comparison of ventricular tachyarrhythmia recurrence between ischemic cardiomyopathy and dilated cardiomyopathy: a retrospective study"

_PeerJ, doi:10.7717/peerj.5312_

## Round 0.1 · original submission · Major Revisions

Your article has received 3 independent peer reviews with unanimous recommendations of 'Major Revisions'. Please address to each of the reviewer's concerns including the following:

1) This article needs language editing by a native English speaker.
2) The methods section is inadequate.
3) The citation to DANISH trial is out of context to the subject of this investigation.
4) Please explain why the number of patients who received ICD over 11 years was only 132?
5) I note that the proportions of patients not receiving guidelines recommended therapies for heart failure are high in both groups. Please comment.
6) Did any patient receive a CRT device?
7) What was the follow-up regimen?

·

Basic reporting

The English language used needs revision to correct grammatical and spelling errors.

In this single centre prospective observational study, Fang et al investigated the prevalence of ventricular tachycardia and ventricular fibrillation in the patients with ischaemic and non-ischaemic cardiomyopathy with ICD in situ for secondary prevention. They also look at the demographic and echocardiographic predictors of VT/VF.

Experimental design

This is a thought-provoking single-centre prospective observational study. The research questioned are well defined but considering the design of the study and the sample size, the results are reports of observations and would not change the practice in the field.

Validity of the findings

Conclusions are not in line with the nature of an observational study.

Additional comments

Major comments:
1. Line 60 of the introduction requires clarification. The authors claim that cardiac arrhythmias are associated with a significant rise in sudden cardiac death. I am assuming the authors were meant to say ventricular tachyarrhythmias instead of cardiac arrhythmias. The most common cardiac arrhythmia, atrial fibrillation, is not associated with significant cardiovascular mortality.
2. Line 65 of the introduction is reemphasising on the beneficiary role of implantable cardioverter-defibrillator as a secondary prevention. In the following sentence, the authors’ emphasis on the improvement in the survival of the patients with heart failure thanks to improved optimal medical therapy makes the reasoning of the authors more supportive of primary prevention ICDs rather than secondary prevention ICD. Definition of primary and secondary prevention ICD needs to be elaborated on in the introduction.
3. If all these ICDs implanted for secondary prevention, the primary presentation of the patients and their presenting rhythm should be presented as part of the study group characteristics.
4. The Methods is very brief and needs elaboration. The authors refer to a separate published study for their methods which appears insufficient. The authors needs to expand this section by explaining their methods in more detail.
5. The diagnostic methods used in the diagnosis of non-ischaemic cardiomyopathy are not described. I would recommend describing the diagnostic methods utilised to rule out ischaemic causes, i.e. coronary angiogram, computed tomography angiogram, myocardial perfusion scan, etc.
6. The primary end of the study requires explanation. It is not clear what the authors meant by recurrence of VT and ICD shock. Was any non-sustained VT considered a primary endpoint? Were inappropriate ICD shocks excluded? What happened to VTs terminated with anti-tachycardia pacing? Were they considered as the primary endpoint of not?
7. The follow up duration is better to be presented in years rather than days. Although the mean follow up is very close to the standard deviation, it is still better to be presented in years.
8. Fig 2 depicted the Kaplan-Meier curve of mortality showing a divergence between ischaemic and non-ischaemic cardiomyopathy starting after 3 years. Considering the small number of the patients with non-ischaemic cardiomyopathy with follow up of 3 years (10 pts) and very small number at 5 years (5 pts), this study is not powered to detect any difference in the mortality rate between ischaemic and non-ischaemic cardiomyopathy and P-value, in this case, would not bear any weight.
9. Line 204 of discussion, refers to the controversies around ICD implantation in non-ischaemic cardiomyopathy by referring to DANISH study (reference number 8). The authors follow by discussing their findings in risk stratification of VT/VF in this study. It needs to be highlighted that DANISH was a primary prevention ICD trial and cannot be referred to in this study as the current study is a prospective observational study of secondary prevention ICD.

Minor Comments:
1. There are multiple grammatical and spelling errors that need to be corrected.
2. Under Echocardiography, line 113, “biplanar” is a typo and should spell biplane.

Reviewer 2 ·

Basic reporting

The authors have employed professional English through out. The introduction of the article has highlighted the importance of ventricular arrhythmia as a cause for sudden death and the improvement achieved with AICD. They have identified a lacuna in the literature which is, a comparison of predictors for recurrent VT in patients with ischemic as opposed to the dilated cardiomyopathy patients.

Experimental design

Fang et al have performed a retrospective analysis of consecutive cardiomyopathy patients who had an AICD implanted as secondary prevention strategy. They classified the patients into ischemic {ICM} and dilated {DCM} with an attempt to highlight the predictors for recurrent ventricular arrhythmia. All events recorded by the device were interrogated and appropriately adjudicated by the investigators. Various parameters were studied including the medications and echocardiographic variables.

Validity of the findings

Appropriate statistical analysis was used, and ROC curves employed to determine the optimal cut off for the echocardiographic parameters.

There was significant difference between the two groups at baseline. The ischemic cardiomyopathy patients were significantly older with higher prevalence of coronary risk factors as compared to those with dilated cardiomyopathy. Additionally, the use of ACE inhibitor was lower in the former. The DCM group had lower EF, larger LV volumes that resulted in increased prevalence of ventricular arrhythmias as compared to the ICM. However, when patients with similar ejection fraction and LV volume were compared between the groups, there was no significant difference in the prevalence of ventricular arrhythmia.

The conclusions are well stated but not discussed sufficiently.

Additional comments

1. Although it is a retrospective study, it sheds light on recurrent ventricular arrhythmia in cardiomyopathy patients. The authors have employed sound statistical methods. However, the discussion of their findings can improve. There is no reasonable explanation given for the findings of increased prevalence of arrhythmia in DCM as compared to the ICM. The fact that when similar ejection fraction and LV volumes were compared, there was no increase in the prevalence of arrhythmia suggest that these are indeed the confounders and probably hearts are more diseased to start with.
2. When the authors described that the patients with “similar LV volume and mean EF were compared”, they did not explain further the variation that was allowed.
3. The VT occurrence is similar at first year. It is not clear when they diverge. It was not obvious from the study if the follow up echocardiogram at this time revealed a larger LV and lower EF compared to the baseline.
4. It is interesting to note that in spite of the greater prevalence of ventricular arrhythmia in DCM patients, the all cause mortality and cardiovascular mortality has not changed. A discussion of this finding was lacking in the manuscript provided.
5. A few studies have been quoted but not linked appropriately to the study during the discussion.
6. The spelling of small – line 236 needs to be corrected.

·

Basic reporting

The authors present a retrospective analysis of predictors for ventricular arrhythmia in patients implanted with a secondary prevention ICD, comparing patients with ischaemic and non-ischaemic cardiomyopathy.The study is single center, inclusion and follow-up are done between 2004-2015. Clinical predictors including cardiac risk factors and medications, echocardiographic parameters including LV function and dimensions were investigated.

The manuscript is well structured and clearly written.

Experimental design

The methods section is very brief referring to the 2016 paper. I would recommend at least mentioning the inclusion criteria, the patient selection (time period of inclusion and follow-up period), the timing and method of data collection including baseline, clinical and echo parameters in relation to follow up (eg was echo from prior to device implantation?) and method of determining endpoint (was there a core-lab analysis of the ICD interrogation to account for inappropriate therapies?)

Statistics
the term 'significant' is used a number of times, it would be better to replace that with 'statistically significant' or omit the term and list the P value for the comparison.
The number of decimal places used should be reevaluated (eg age just use one decimal point, and for the standard deviation don't use a decimal).
Median with confidence interval is more appropriate than mean with SD for non-normal distributed parameters (that should be apparent from the test for normal distribution)
In one of the regression models both LV diameter and LV diameter >158ml were used, please check with your statistician if it is appropriate to include both at the same time in the regression model.

Validity of the findings

See also the comments in 2.

There is continuity in structure regarding research question, study design, results, discussion and conclusions.

Additional comments

32: consider listing follow-up in months or years instead of days
87: as previously mentioned, the methods should be described in this paper as well
102: which hospital?
103: how was coronary disease ruled out (angiography, CT)?
111: when was echo performed and analyzed (at inclusion to registry prior to device implantation, were all patients on 3 months of maximally tolerated heart failure medication?)
116: what duration of VF/VT was considered significant? How were events evaluated for inappropriate device therapy? Was a core-lab analysis method used with independent reviewers?
119: did all cause mortality only include the listed causes or also others (eg death from any cause)
122: see comments with statistics.
140: please correct 'valve'
146, 149,150,161, 166,174 etc see comments regarding significance
173 see comments with statistics
229: is lack of ACE the reason for increased VT/VF or is it a marker of decreased renal function or increased disease severity and hypotension that do not allow for addition of ACE?

Figures & tables: please consider the comments regarding decimal point, and perhaps use months rather than days.

---

## Round 0.2 · Minor Revisions

Your revised work has been reviewed and we have found improvements, but the following has prevented acceptance.
1. Usage of English language remains unacceptable.
2. Methods section remains inadequate.

·

Basic reporting

This article is revised and reads better now with less grammatical errors.

Experimental design

This is a retrospective, single centre, and observational study. In limitations of small observational studies, this study is well designed and well conducted. The endpoints are clear and as far as one could ascertain from the article, the follow-up and data collection was performed meticulously.

Validity of the findings

In limitations of a small single centre study, data is well presented and analysed. The conclusion is linked to the original research question

Additional comments

The manuscript is nicely revised and is more comprehensible, however, it could benefit from another revision. The language and grammar could benefit from another revision.

---

## Round 0.3 · accepted · Accept

I have corrected several residual errors in the manuscript on your behalf. PeerJ staff will send you the original Word doc.